# Antipsychotic Drug-Mediated Adverse Effects on Rat Testicles May Be Caused by Altered Redox and Hormonal Homeostasis

**DOI:** 10.3390/ijms232213698

**Published:** 2022-11-08

**Authors:** Aleksandra Nikolić-Kokić, Nikola Tatalović, Jelena Brkljačić, Milica Mijović, Vojkan Nestorović, Ana Mijušković, Zorana Oreščanin-Dušić, Teodora Vidonja Uzelac, Milan Nikolić, Snežana Spasić, Duško Blagojević, Čedo Miljević

**Affiliations:** 1Department of Physiology, Institute for Biological Research “Siniša Stanković”—National Institute of Republic of Serbia, University of Belgrade, 11000 Belgrade, Serbia; 2Department of Biochemistry, Institute for Biological Research “Siniša Stanković”—National Institute of Republic of Serbia, University of Belgrade, 11000 Belgrade, Serbia; 3Institute of Pathology, Faculty of Medicine, University of Priština, 38220 Kosovska Mitrovica, Serbia; 4Institute of Physiology, Faculty of Medicine, University of Priština, 38220 Kosovska Mitrovica, Serbia; 5Department of Biochemistry, Faculty of Chemistry, University of Belgrade, 11000 Belgrade, Serbia; 6Department of Chemistry, Institute of Chemistry, Technology and Metallurgy–National Institute of Republic of Serbia, University of Belgrade, 11000 Belgrade, Serbia; 7Outpatient Department, Institute of Mental Health, School of Medicine, University of Belgrade, 11000 Belgrade, Serbia

**Keywords:** testicles, prostate, atypical antipsychotics, antioxidant enzymes

## Abstract

Sexual dysfunction, as a noticeable adverse effect of atypical antipsychotic drugs (APDs) for the treatment of schizophrenia, has not been investigated in detail. A study was undertaken to investigate whether 28-day long treatment with clozapine, ziprasidone or sertindole (using a recommended daily dose for atypical antipsychotic therapy), induced histopathological changes both in rat testicles and prostate, changed the activity of the antioxidant defence system and altered blood testosterone and prolactin. Clozapine, ziprasidone and sertindole induced histopathological changes in rat testicular tissue, which could be attributed to a disturbed testicular antioxidant defence system in addition to an altered prolactin to testosterone ratio. None of the APD treatments induced histopathological changes in prostate. Our results demonstrate that APDs have the capacity to change both redox and endocrinological balance. One or both outcomes could underline testicular degeneration and disturbed spermatogenesis.

## 1. Introduction

Atypical antipsychotic drugs (APDs) are widely used for the treatment of schizophrenia and other psychiatric disorders, since they are associated with fewer side effects compared to those caused by typical antipsychotics (extrapyramidal symptoms) [1]. However, APDs can increase the risk of metabolic disorders since many of them are associated, to variable degrees, with weight gain, hyperglycemia, hypertension, metabolic alterations including adverse effects on lipid and glucose metabolism, changes in the heart, kidneys and the liver [2,3,4,5,6].

Some typical and atypical antipsychotics also have the potential to negatively affect sexual function [7] and hence the quality of life of patients. Typical neuroleptics affect male sexual performance (by inducing anorgasmia, decreased libido, ejaculation and erectile dysfunction). In addition, APDs also pose a risk to reproductive health [8]. Patients that received second-generation antipsychotics, such as olanzapine and clozapine, experienced reproductive dysfunction [9]. Clozapine and haloperidol affect hormonal status and the process of spermatogenesis in rats [10]. While some studies reported that “atypical” neuroleptics barely raised prolactin [11,12], others reported increased prolactin in patients treated with olanzapine [13] or with risperidone [14]. De Rosa and co-workers pointed out that hyper-prolactinaemia was related to hypogonadism, through inhibition of gonadotropin-releasing-hormone, follicle stimulating hormone, luteinizing hormone and testosterone secretion, and disturbed spermatogenesis, reduced sperm motility, semen quality and morphological changes in the testicles [11]. It should be noted that in most cases the effects of drugs are reversed after their intake is paused [15]. However, schizophrenia is a chronic disorder and requires a lifetime of treatment [16]. Therefore, studies focused on understanding the molecular mechanisms underlying APDs’ adverse effects on the reproductive system should contribute to developing novel and better treatment strategies for patient treatment.

Oxidative stress and the concomitant production of reactive oxygen species (ROS) has been implicated in male infertility [17]. However, the link between the oxidative stress, antipsychotics and sexual disorders/male infertility is missing. Our previous results showed that treatment with APDs led to histopathological changes in the liver, heart and kidneys and were accompanied by changes in the antioxidant enzyme defence system [18,19,20]. Clozapine, ziprasidone and sertindole caused different degrees of adverse effects in the examined tissues and on antioxidant enzyme activity, suggesting different modes of action including oxidative stress.

Taking into account the adverse effects of APDs on reproductive function and tissue morphology, the aim of our current investigation was to evaluate whether clozapine, ziprasidone and sertindole induced histopathological changes in the testicles and prostate, and to explore their impact on the tissue antioxidant defence system.

## 2. Results

### 2.1. Relative Testicle and Prostate Weights

Analysis of variance showed significant differences in relative testicular mass but the *post-hoc* test did not yield significant differences between individual groups (Figure 1). Sertindole-treated rats had a higher relative prostate mass than clozapine- and ziprasidone-treated rats (*p* < 0.05, both), but there was no significant difference compared to the control group (Figure 1). Higher relative prostate mass in the sertindole group was not a consequence of decreased body weight caused by treatment and a consequential increase in the relative mass of visceral organs, since there were no significant differences in body weight among the groups at the end of the experiment (Figure 1).

In addition, in three sertindole-treated rats, their prostates were visibly enlarged at the point of necropsy. One of these rats (the one with the highest relative prostate mass in the entire experiment) was also the one with the lowest relative testicular weight in the whole experiment. None of the treatments had statistically significant effect on the activity of alanine aminotransferase (ALT) and aspartate aminotransferase (AST) in the plasma (Table 1).

### 2.2. Histopathological Examination of the Testicular and Prostate Tissue

There were no significant differences between groups with respect to colour, shape and texture of testicles and prostates. Histopathological examination of testicles (Table 2) from the control group revealed normal seminiferous epithelium and complete spermatogenesis with many spermatozoa present (Johnsen score 10) (Figure 2A,B).

In 4 clozapine-treated rats (67%) less than five spermatozoa per tubule and a few late spermatids (Johnsen score 8) were found (Figure 2C,D). Disorganised seminiferous epithelium and slightly impaired spermatogenesis with many late spermatids (Johnsen score 9) were observed in 2 clozapine-treated rats (33%) (Figure 2E,F). Johnsen score 8 (Figure 2G,H) and 9 (Figure 2I,J) were equally represented in each of 3 (50%) animals in the ziprasidone-treated group.

Sertindole-treated rats showed seminiferous tubules with few germinal cells and pyknotic nuclei and extensive disorganization (Johnsen score 6) (Figure 2K,L) in 2 (25%) rats and seminiferous tubule with no late spermatids, with many early spermatids (Johnsen score 7) (Figure 2M,N) in 6 (75%) rats.

Histopathological examination of prostates showed normal glands/fibromuscular stroma ratio, without any sign of benign prostate hyperplasia in control (Figure 3A) as well as in clozapine- (Figure 3B), ziprasidone- (Figure 3C) and sertindole-treated rats (Figure 3D).

There was no evidence of apoptosis in routine testicular slides stained by hematoxylin eosin (HE).

### 2.3. Activity of Antioxidant Enzymes in Testicles and Prostate

Testicles: A decrease in catalase (CAT) activity (*p* < 0.001) and an increase in copper-zinc superoxide dismutase (SOD1) activity (*p* < 0.05) were detected in clozapine-treated rats (Figure 4). Both ziprasidone and sertindole decreased both CAT activity (*p* < 0.001) and selenium–glutathione peroxidase (GPx) activity (*p* < 0.05) (Figure 4). Decreased glutathione reductase (GR) activity was detected only in ziprasidone-treated rats.

Prostates: In clozapine-treated rats increases in SOD1 activity (*p* < 0.01), GPx activity (*p* < 0.05), CAT activity (*p* < 0.01) and GR activity (*p* < 0.05) were observed. Ziprasidone treatment increased the activity of both SOD1 (*p* < 0.001) and GR (*p* < 0.01). There were no statistically significant changes in the activities of antioxidant enzymes in prostates of sertindole-treated rats (Figure 5).

Correlation analysis showed that the Johnsen score was highly and positively correlated with CAT activity in testicles (r = 0.807, *p* < 0.0001), less so with activity of GPx (r = 0.485, *p* < 0.01). In contrast, Johnsen score did not correlate with the activity of antioxidant enzymes in prostate. Relative prostate weight negatively correlated with the activities of both SOD1 and GR in prostate (r = −0.763, *p* < 0.001 and r = −0.620, *p* < 0.05, respectively).

### 2.4. Hormone Levels in Plasma

There were no statistically significant changes in the mean plasma prolactin and testosterone concentrations between treatment groups, (Figure 6), probably due to large variability within the groups. However, the ratio of prolactin to testosterone was significantly higher in the sertindole-treated group compared to the control- and clozapine-treated groups (*p* < 0.05, both, Figure 6).

## 3. Discussion

Although APDs have been shown to have a more favorable side effect profile than typical antipsychotics, they still pose a risk to metabolic and reproductive health. The results of our current study show that clozapine, sertindole and ziprasidone caused changes in the seminiferous epithelium and affected stages of spermatogenesis. According to Ardiç and coworkers (2021) olanzapine also caused histopathological changes in rat testes [21]. The authors found disorganisation within seminiferous tubules (swelling of Leydig cells and intracytoplasmic vacuolisation of Sertoli’s cells in testicular parenchyma), exfoliation and degeneration of germ cells. Olanzapine affected sperm morphology but did not affect sperm motility [21].

Spermatogenesis, the most prolific replicative process, demands the highest rate of oxygen consumption and consequently produces high levels of reactive oxygen species (ROS). ROS have a dual role in biological systems, both beneficial [22] and harmful [23] depending on their nature, concentration, location and exposure time [24,25,26,27]. A decrease in antioxidant activity in semen is associated with idiopathic infertility [28].

In our current study, all three APDs disturbed the antioxidant defence system in testicular tissue. Decrease testicular CAT activity, observed in all experimental groups, along with reduced GPx activity (in ziprasidone and sertindole-treated rats) suggests reduced peroxides elimination in the testicles. Additionally, clozapine treatment also led to increased SOD1 activity, suggesting faster elimination of superoxide anion and simultaneously increased generation of hydrogen peroxide. Decreased CAT and GPx activities directly correlated with the condition of the seminiferous tubules (revealed by the Johnsen score) suggesting that the hydrogen peroxide accumulated in cells might be involved in testicular tissue damage. Ziprasidone decreased the activity of GR, responsible for glutathione turnover and preservation of the cellular glutathione pool. Lower glutathione turnover can lead to less glutathione availability for spermatogenesis. This could additionally contribute to testicular degeneration and infertility. Lower GR activity could be also a consequence of lower peroxide elimination by GPx and no requirement for oxidised glutathione reduction in a way to preserve the NADPH cellular pool. Other antipsychotics also impair antioxidant enzyme activity. Olanzapine increased SOD activity whilst decreasing glutathione in rat testicles, related to abnormal sperm morphology and histopathological changes [21]. Risperidone also decreased glutathione, reduced semen quality and induced histopathological changes [29]. Although in the testicles of sertindole-treated rats “only” lower CAT and GPx activities were measured compared to ziprasidone and closapine treatment (suggesting a lower oxidative stress burden), a higher adverse effect on spermatogenesis occurred. An increased prolactin/testosterone ratio observed after sertindole treatment was another factor that resulted in poor spermatogenesis.

Previous studies have shown that risperidone induced reproductive toxicity in male rats target Leydig cells and the hypothalamic-pituitary-gonadal axis by inducing oxidative stress. Olanzapine and risperidone were also found to decrease plasma testosterone in rats [27]. In cocaine-induced reproductive dysfunction in male mice, reduced hypothalamic gonadotropin-releasing-hormone expression and plasma testosterone were noted. Moreover, the GPx-1 gene has been shown to antagonise this dysfunction, suggesting peroxide involvement [28]. It is known that ROS can reduce male sex hormones and disturb the hormonal balance that regulates reproductive function [29]. Accordingly, by interfering with the normal release of hormones, ROS interferes with these basic reproductive functions [30]. Our results demonstrate that both hormonal and redox disturbance contribute to the adverse effects of sertindole in testicles. The relationship between APDs and sexual dysfunction is mediated in part by antipsychotic blockade of pituitary dopamine D2 receptors causing increased prolactin secretion. However, direct correlation has not been established between raised prolactin and clinical symptoms [31].

Sertindole caused the most prominent histopathological changes in the testicles along with impaired hypothalamic-pituitary and gonadal axis function. The prolactin/testosterone ratio proved to be a useful tool [32]. Hyper-prolactinaemia accompanied by decreased testosterone was reported in olanzapine-treated male rats [21,33]. Degeneration of Leydig cells (the focus of testicles androgen biosynthesis in adult rats) is usually accompanied by decreased testosterone [34]. A significant decrease in circulating testosterone may be caused by oxidative stress in the testes resulting in decreased testosterone production, as a result of injury to Leidig cells or other endocrine structures such as the anterior pituitary [35]. Steroidogenesis also generates ROS, which are mainly produced by mitochondrial respiration and catalytic reactions of steroidogenic cytochrome P450 enzymes [36]. ROS generated in this way have been identified as inhibiting subsequent steroid production and damaging sperm mitochondrial membranes [37]. Our results show the presence of many early spermatids, suggesting that oxidative imbalance may be associated with increased immature sperm count, probably an indirect effect on male hormone production associated with spermatogenesis [38]. In our study, redox disturbances observed in sertindole-treated rats paralleled discrete changes in prolactin and testosterone concentrations that resulted in the increased prolactin/testosterone ratio. At the same time, the disturbed ratio had no effect on prostate antioxidative enzymes. However, negative correlation between prostate relative weight and the activities of SOD and GR suggests the importance of superoxide elimination and preservation of redox state for prostate health. Our results show that 4-week long treatment with clozapine, ziprasidone or sertindole had no effect on prostate morphology as normal glands/fibromuscular stroma ratio without any sign of benign prostate hyperplasia was observed. At the same time treatment with ziprasidone led to an increase in the activities of SOD1 and GR in prostate. Clozapine treatment also raised the activities of SOD1, GR, GPx and CAT (Figure 5), suggesting faster elimination of superoxide anion and hydrogen peroxide as well as higher glutathione turnover. We can assume that the prostate, due to enhanced capacity of the antioxidant system, has the ability to maintain redox homeostasis, regulate ROS concentration within non-toxic homeostatic levels and protect this tissue from oxidative damage. However, it seems that prolonged treatment with APDs would ultimately overwhelm the capacity of antioxidant system leading to oxidative stress and tissue damage, thus affecting the organ function, despite a relative increase in the mass of prostate of sertindole-treated rats compared to closapine and ziprasidone treatments.

In human studies, it is extremely difficult to distinguish adverse effects that can be attributed to a psychotic illness from those that can be attributed to the use of a medication. The results of our current study confirmed the proposed causative role of oxidative disbalance in APD-induced disruption of reproductive function in males and showed that APD-induced oxidative stress alongside endocrinological changes can lead to testicular degeneration, disturbed spermatogenesis and ultimately male subfertility. Understanding this relationship in more detail will help future management of patients and will improve the current therapies.

## 4. Materials and Methods

### 4.1. Chemicals

Ziprasidone (Zeldox) was from Pfizer (Vienna, Austria), clozapine was from Alvogen (Remedica Ltd., Limassol, Cyprus) and sertindole (Serdolect) was from H. Lundbeck (Valby, Denmark).

### 4.2. Animals and Drug Treatment

All procedures complied with directive 2010/63/EU regarding the protection of animals used for experimental and other scientific purposes. Thirty-two healthy adult male Wistar albino rats (three months old, weight 300–350 g) were randomly divided into four experimental groups with eight rats in each. Rats were kept under standard conditions at 22 °C with a 12 h light/dark cycle, provided with food (rodent laboratory chow produced by Veterinarski Zavod Subotica, Subotica, Serbia) and drinking water ad libitum.

All drugs were prepared (water suspension of pulverised tablets) and administered daily in the morning via a gastric tube to ensure that no drug loss occurred. Rats were dosed according to the drug calculation formula [39]. Rats were exposed to water (control group), clozapine (45 mg/kg/day), ziprasidone (20 mg/kg/day) or sertindole (2.5 mg/kg/day) for four weeks. Following overnight fast, rats were euthanised by rapid decapitation

### 4.3. Blood and Tissue Collection

Blood was collected immediately after decapitation in tubes coated with heparin (500 I.U. per mL of blood). Plasma and erythrocytes were separated by centrifugation (MiniSpin, Eppendorf, 3000× *g*, 10 min). Plasma was stored at −20 °C for the measurement of prolactin, testosterone ALT and AST. The testicles and prostate were excised immediately. Colour, shape and texture of testicles and prostate were observed. One testicle was frozen in liquid nitrogen and kept at −80 °C until further analysis while the remaining testicle was used for histopathological analysis. Prostates from one half of the rats in each experimental group were also frozen in liquid nitrogen and kept at −80 °C, while the prostates of the remaining rats were used for histopathological analysis.

### 4.4. Measurement of Prolactin, Testosterone, ALT and AST

They were measured in blood plasma using immunoassay analysers based on electrochemical luminescence detection technology. Prolactin was measured using the Immulite 2000 Immunoassay System (Siemens Healthcare GmbH, Erlangen, Germany) while testosterone was measured using the Cobas e 411 analyser (Roche Diagnostics GmbH, Mannheim, Germany). For each rat the prolactin to testosterone ratio was calculated. Activities of ALT and AST in the plasma were measured by COBAS INTEGRA 400 plus (Roche Diagnostic GmbH, Mannheim, Germany) automated analyser (Roche Diagnostics, Mannheim, Germany).

### 4.5. Tissue Preparation and Determination of Antioxidant Enzyme Activities

For preparation of whole tissue extracts the tissues were homogenised in 10 volumes (wt/vol) of 50 mM Tris-HCl, 0.25 M sucrose, 1 mM EDTA, pH 7.4 and sonicated 3 × 10 sec at 10 MHz (Sonopuls, Bandelin, Berlin, Germany) on ice followed by 60 min of centrifugation at 4 °C and 105,000× *g* (Beckman L7-55 Ultracentrifuge). The supernatants were used as whole tissue extracts.

Total SOD activity was determined by the adrenaline method [40]. One SOD unit was defined as the amount of enzyme necessary to decrease the rate of adrenalin auto-oxidation by 50% at pH 10.2. For the determination of SOD2 activity, the assay was performed after pre-incubation with 4 mM potassium cyanide. SOD1 activity was calculated as the difference between the total SOD and SOD2 activities. CAT activity was determined according to Beutler (1982) [41]. One unit of CAT activity was defined as the amount of enzyme that decomposes 1 mmol H_2_O_2_ per minute at 25 °C and pH 7.0. The activity of GPx was determined by glutathione reduction of t-butyl hydroperoxide using a modification of the assay described by Paglia and Valentine (1967) [42]. One unit of GPx activity was defined as the amount of enzyme needed to oxidise 1 µmol NADPH per min at 25 °C and pH 7.0. GR activity was determined using the method of Glatzle and colleagues (1974) [43]. One unit of GR activity was defined as the amount of enzyme needed to oxidise 1 µmol NADPH per min at 25 °C and pH 7.4. All enzyme activities are expressed as units (U) per mg of protein. The protein concentration was determined by the Lowry method [44], using bovine serum albumin as a standard.

### 4.6. Histopathological Examination

Testicles and prostates were fixed in 4% neutral buffered paraformaldehyde solution for 24 h, followed by dehydration in increasing concentrations of ethanol and xylene and then embedded in Histowax (Histolab Product AB, Askim, Sweden). A rotary microtome HistoCore AUTOCUT (Leica Biosystems, Wetzlar, Germany) was used to obtain 5 μm thick slices. Hematoxylin-eosin (H&E) staining [45] was used for testicular and prostate tissue visualisation. Microscope slides were examined using light microscopy Leica DM LS2 (Leica Microsystems, Wetzlar, Germany) while original photographs were made using Canon Power Shot S70 camera (Canon U.S.A. Inc., Huntington, NY, USA).

### 4.7. Histopathological Examination of Spermatogenesis (Visualisation of the Spermatogenic Cells)

Testicles of control group n = 7 were taken for histopathological examination; clozapine group n = 6; ziprasidone group n = 6 and sertindole group n = 8 (Table 2). All tubular segments in each section of the testicle were evaluated systematically with special reference to the quality of epithelium of seminiferous tubules. Damaged tubules at the edges of the sections were excluded. A sample was considered satisfactory if at least 30 seminiferous tubules were visible for cell counting. The seminiferous tubules were graded using the Johnsen score as: Score 10 (complete spermatogenesis with regular tubules and many spermatozoa); score 9 (slightly impaired spermatogenesis with many late spermatids, disorganised epithelium); score 8 (less than five spermatozoa per tubule, a few late spermatids); score 7 (no late spermatids, many early spermatids); score 6 (no late spermatids, few early spermatids); score 5 (no spermatids, many spermatocytes); score 4 (no spermatids, few spermatocytes); score 3 (presence of spermatogonia only); score 2 (presence of Sertoli’s cells only); or score 1 (no seminiferous epithelium) (Johnsen 1970). The scoring was performed at 200× and 400× field magnification.

### 4.8. Statistical Analyses

Statistical analyses were performed according to protocols described by Hinkle et al. (1994) [46]. Differences between groups from Johnsen scoring were tested by Pearson’s chi-square test (χ2). Differences in plasma prolactin and testosterone and prolactin to testosterone ratio, as well as in the activity of antioxidant enzymes in both testicles and prostate were tested by one–way analysis of variance (ANOVA) followed by Tukey’s multiple comparison post-hoc test. Correlations between hormone levels as well as Johnsen scores with activity of antioxidant enzymes in testicles and prostates were calculated using Spearman’s correlation coefficient. A probability level of *p* < 0.05 was considered statistically significant. All analyses were conducted using GraphPad Prism 8 (GraphPad Software Inc., San Diego, CA, USA).

## 5. Conclusions

Clozapine, sertindole and ziprasidone treatments induced histopathological changes in rat testicles which coincided with disruption to the antioxidant defence system. Sertindole treatment induced the most prolific histopathological changes in testicles and increased the prolactin–testosterone ratio, which most likely caused further testicular tissue damage. None of the above-mentioned treatments induced histopathological changes in the prostate.

## Figures and Tables

**Figure 1 ijms-23-13698-f001:**
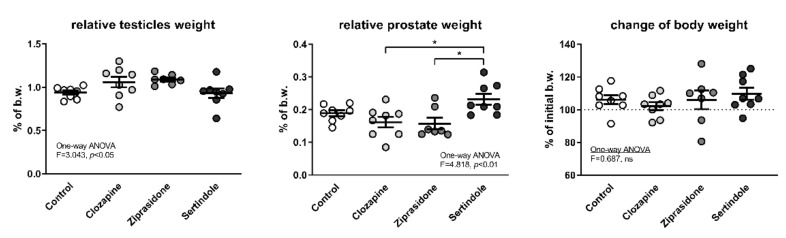
Relative testicle and prostate weight at the end of the experiment together with the change in body weight during the four-week treatment with APDs. Mean values and standard errors are presented; b.w.—body weight; *—*p* < 0.05.

**Figure 2 ijms-23-13698-f002:**
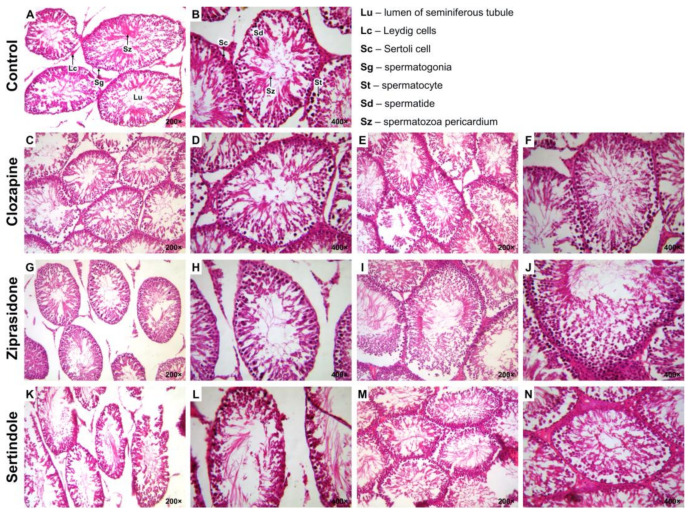
Photomicrographs of rat testicular tissue structure, H&E stain. Control group–(**A**) and (**B**): normal seminiferous tubular morphology (Johnsen score 10); Lu–lumen of seminiferous tubule, Lc–Leydig cells, Sc–Sertoli’s cell, Sg–spermatogonia, St–spermatocyte, Sd–spermatid, Sz–spermatozoa ((**A**): 200×; (**B**): 400×). Clozapine-treated group–(**C**) and (**D**): less than five spermatozoa per tubule, a few late spermatids (Johnsen score 8); (**E**) and (**F**): slightly impaired spermatogenesis with many late spermatids and disorganised epithelium (Johnsen score 9) ((**C**) and (**E**): 200×; (**D**) and (**F**): 400×). Ziprasidone-treated group–(**G**) and (**H**): less than five spermatozoa per tubule, a few late spermatids (Johnsen score 8); (**I**) and (**J**): slightly impaired spermatogenesis with many late spermatids and disorganised epithelium (Johnsen score 9) ((**E**) and (**G**): 200×; (**F**) and (**H**): 400×). Sertindole-treated group– (**K**) and (**L**): seminiferous tubule with few germinal cells and pyknotic nuclei and extensive disorganisation (Johnsen score 6); (**M**) and (**N**): seminiferous tubule with no late spermatids, many early spermatids (Johnsen score 7) ((**K**) and (**M**): 200×; (**L**) and (**N**): 400×).

**Figure 3 ijms-23-13698-f003:**
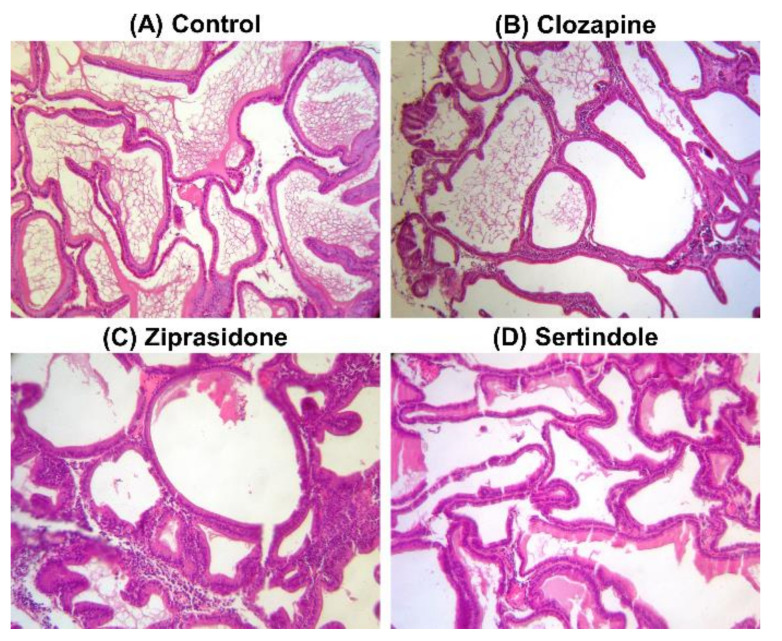
Photomicrographs of rat prostate tissue structure, H&E stain, 200×. Control group–(**A**); Clozapine-treated group–(**B**); Ziprasidone-treated group–(**C**); Sertindole-treated group–(**D**).

**Figure 4 ijms-23-13698-f004:**
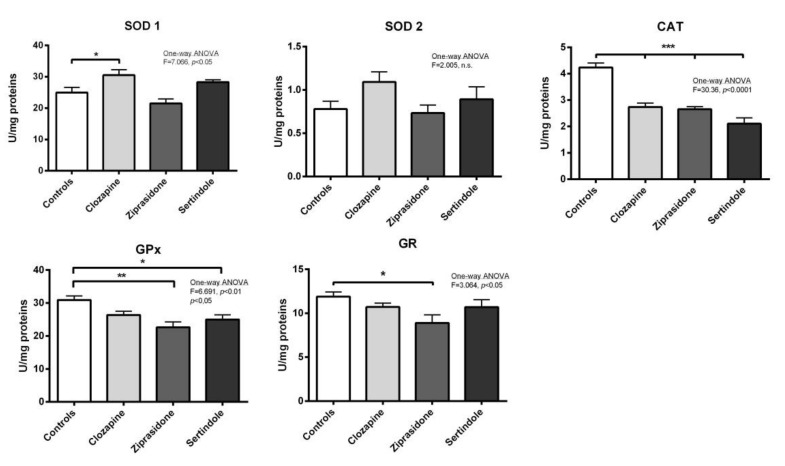
Activities of testicular antioxidant defence enzymes in rats after treatment with clozapine, ziprasidone and sertindole. SOD1, SOD2, CAT, GPx and GR activities were measured in control- and clozapine-, ziprasidone-and sertindole-treated rats. The values represent the means ± SEM. Statistical significance of the difference between experimental groups (one-way ANOVA): * *p* < 0.05, ** *p* < 0.001 and *** *p* < 0.001.

**Figure 5 ijms-23-13698-f005:**
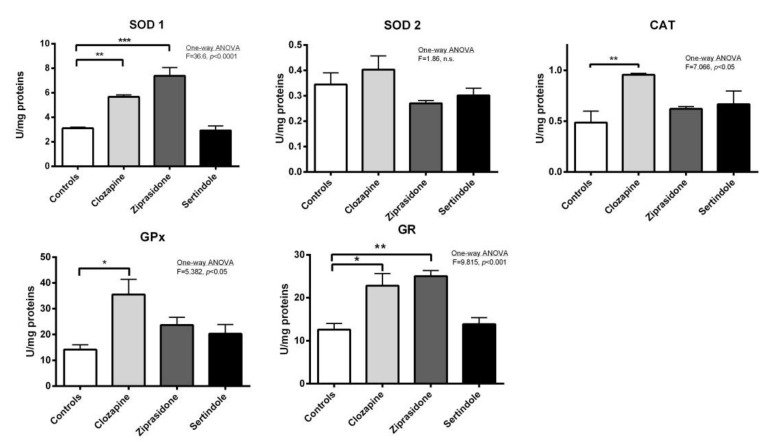
Activities of prostate antioxidant defence enzymes in rats after treatment with clozapine, ziprasidone and sertindole. SOD1, SOD2, CAT, GPx and GR activities were measured in control- and clozapine-, ziprasidone- and sertindole-treated rats. The values represent the means ± SEM. Statistical significance of the difference between experimental groups (one-way ANOVA): * *p* < 0.05, ** *p* < 0.001 and *** *p* < 0.001.

**Figure 6 ijms-23-13698-f006:**
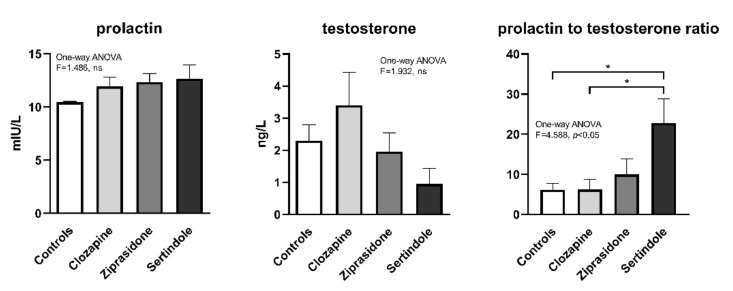
Concentration of prolactin and testosterone after 28 days of treatment with clozapine, ziprasidone and sertindole. * *p* < 0.05.

**Table 1 ijms-23-13698-t001:** The alanine aminotransferase (ALT) and aspartate aminotransferase (AST) activities (U/L) in rats after treatment with clozapine, ziprasidone and sertindole.

	Control	Clozapine	Ziprasidone	Sertindole
Plasma ALT (U/L)	82.8 ± 6.4	69.6 ± 3.1	88.8 ± 10.2	72.1 ± 12.6
Plasma AST (U/L)	276 ± 21.5	244.8 ± 21.09	261.2 ± 29.2	258.2 ± 16.8

**Table 2 ijms-23-13698-t002:** Johnsen score in rat testicles after 4-weeks of treatment with APDs. Effects of treatments were tested by Pearson’s chi-squared test.

		Johnsen Score										
		1	2	3	4	5	6	7	8	9	10	
Control	N										7	n = 7
	%										100	
Clozapine	n								4	2		n = 6
	%								67	33		
Ziprasidone	n								3	3		n = 6
	%								50	50		
Sertindole	n						2	6				n = 8
	%						25	75				
		χ2 = 57.77; *p* < 0.001										

## Data Availability

The data used to support the findings of this study are included within the article.

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
