# Peer review of "Antipsychotic Drug-Mediated Adverse Effects on Rat Testicles May Be Caused by Altered Redox and Hormonal Homeostasis"

_ijms, 2022, doi:10.3390/ijms232213698_

Round 1
Reviewer 1 Report
Introduction
English language and style needs editing
Disordered and repetitive introduction
Abbreviation APD – stands for what?
Results
Results relating to Figure 1. The text states that there is a difference in body weight between the groups but there is no significant difference in the graph 1C.
There is no significant difference in testicular weights between the groups so the detail about individual testicular size is irrelevant.
Prostate weights are significantly different for sertindole compared to the other drug treatments, why was the comparison not made/mentioned for sertindole vs control?
The legend says 6 week treatment whereas materials and methods says 4 weeks.
b.w is not used as an abbreviation in this figure.
What is F? as in F= 3.043 in the figure
Results table 1. I do not understand this table at all. The n numbers would suggest that not all rats were examined, with n=7 or n=6 stated for 3 of 4 groups, this is not mentioned in the methods section.
In the clozapine treated animals 4/6 are said to have Johnsen score of 8 and this is stated to be 77.7% (in text) and 78% in table. Surely 4/6 = 66.67% of rats.
Results section Activity of Antioxidant Enzymes in Testicles and Prostate
What is GR?
Testicle results are reported before this section and should be moved. Graphical representation is said to be in figure 3 but is actually figure 4, with figures out of sync from here on. Results are not reported systematically, should be reported in more detail.
For GPX what are the authors measuring? There are eight GPX proteins, which are expressed to differing degrees in different tissues e.g. GPX4 highly expressed in testes, but lower in prostate, GPX5 and 6 mostly expressed in epididymis. They are also seen if different subcellular compartments, some of which would be relevant to mitochondrial antioxidant protection others not.
Prostate results are figure 5 not figure 4.
Negative correlation of prostate weight and SOD1/GR – how was this calculated? There is no sig diff in prostate weight noted between control group and clozapine or ziprasidone treated group and there is no sig diff in SOD1 and GR between control and sertindole so I find it hard to comprehend how this can be stated.
The authors state “The mean plasma prolactin and testosterone concentrations differed between treatment groups, but without statistical significance” however if there is not statistical significance this cannot be said to be so.
Does prolactin/testosterone ratio correlate with prostate weight? This would seem the most striking finding in sertindole group
Could the authors test reproductive capacity in these rats? i.e did the changes in spermatogenesis reduce ability to father offspring?
Discussion
Over interpretation of results, for example the assumption of increased hydrogen peroxide causing damage to the tissue where the authors have not shown that H2O2 is increased.
Overall I am not convinced they show that ROS is what is causing the phenotype in these rats, further confirmatory studies/assays would need to be carried out. Longer term treatment of the rats perhaps, fertility studies and proof that ROS is, in fact, increased in these tissues.
Johnson or Johnsen, which is correct?
Materials and methods
Is water an appropriate control? What is the formulation of the tablets? Presumably there are other components than just the drug which need to be accounted for in the control rats.
For antioxidant assays was the tissue extract supernatant used without any further modifications?
Author Response
Introduction
English language and style needs editing
Disordered and repetitive introduction
Abbreviation APD – stands for what?
Answer: Thanks for the critique the introduction has been rewritten:
Atypical antipsychotic drugs (APDs) are widely used for the treatment of schizophre-nia and other psychiatric disorders, since they are associated with fewer accompany-ing side effects as compared to those caused by typical antipsychotics (extrapyrami-dal symptoms). However, APDs can increase a risk for metabolic disorders since many of them are associated, to variable degrees, with weight gain, hyperglycemia, hypertension, metabolic alterations including adverse effects on lipid and glucose metabolism, changes in the heart, kidneys, and the liver [2-6]. Some typical and atypical antipsychotics also have the potential to negatively affect sexual function [7] and hence the quality of life of patients. Typical neuroleptics were shown to affect male sexual performance (by inducing anorgasmia, decreased libido, ejaculation and erectile dysfunction), however, it was shown that APD medications also pose a risk to reproductive health [8]. Namely, it was reported that patients receiving second-generation antipsychotics, such as olanzapine and clozapine, experience reproductive dysfunction [9]. Clozapine and haloperidol have been shown to affect hormonal status and the process of spermatogenesis in rats [10]. While some studies reported that “atypical” neuroleptics barely raise prolactin [11, 12], others reported increased prolactin level in patients treated with olanzapine [13] or risperidone [14]. De Rosa and co-workers pointed out that hyperprolactinaemia was related to hypogonadism, through inhibition of gonadotropin-releasing-hormon, follicle stimulating hormone, luteinizing hormone and testosterone secretion, and disturbed spermatogenesis, reduced sperm motility, semen quality, and morphological changes in the testis [11]. It should be noted that in most cases the effects of drugs are reversed after intake is interrupted [15]. However, schizophrenia as a chronic disorder requires a lifetime of treatment [16]. Therefore, studies focused on understanding the molecular mechanisms underlying APDs adverse effects on the reproductive system should contribute to developing novel and better treatment strategies. These should achieve equal therapeutic response but without affecting male reproductive function thus improving men’s quality of life.
Oxidative stress and the concomitant production of reactive oxygen species (ROS) have been implicated in male infertility and may be a common pathological pathway associated with it [17]. However, link between the oxidative stress in male infertility and antipsychotics treatment caused sexual disorders is missing. Our previous results showed that treatment with APDs led to histopathological changes in liver, heart and kidneys that were accompanied with changes in antioxidant enzyme defence [18, 19, 20]. Clozapine, ziprasidone and sertindole caused different degrees of adverse effects in the examined tissues and on antioxidant enzyme activity suggesting different modes of action including oxidative stress.
Taking into account the adverse effects of APDs on reproductive function and tissue morphology, the aim of our investigation was to evaluate whether clozapine, ziprasidone and sertindole induce histopathological changes in the testis and prostate, and explore their impact on the tissue antioxidant defence system.
APD is singular of plural shortcut for AntiPsychotic Drugs (APDs)
Results
Results relating to Figure 1. The text states that there is a difference in body weight between the groups but there is no significant difference in the graph 1C.
There is no significant difference in testicular weights between the groups so the detail about individual testicular size is irrelevant.
Prostate weights are significantly different for sertindole compared to the other drug treatments, why was the comparison not made/mentioned for sertindole vs control?
Answer: That part of results has been rewritten:
Analysis of variance showed significant differences in relative testicular mass but the post-hoc test did not yield significant differences between individual groups (Figure 1.). Sertindole-treated rats had a higher relative prostate mass than clozapine- and ziprasidone-treated (p<0.05, both), while there was no significant difference compared to control group (Figure 1.). Higher relative prostate mass in the sertindole group was not a consequence of a decrease in body weight caused by treatment and a consequential increase in the relative mass of visceral organs, since there were no significant differences in body weight among the groups at the end of the experiment (Figure 1.). There was an increase in the mean body weight during the experiment in all experimental groups. In addition, in three sertindole-treated rats their prostates were visibly enlarged at the point of necropsy. One of these rats (the one with the highest relative prostate mass in the entire experiment) was also the one with the lowest relative testicular weight in the whole experiment.
The legend says 6 week treatment whereas materials and methods says 4 weeks.
Answer: Thanks for noticing our mistake now changed in 4weeks
b.w is not used as an abbreviation in this figure.
Answer: Abbreviation b.w. have been used in Y axis title of all three graphs in Figure 1 (% of b.w., % of b.w. and % of initial b.w.), that is the reason it was introduced in the legend of the Figure 1.
What is F? as in F= 3.043 in the figure
Answer: The F-statistic is a ratio of two variances. In one-way ANOVA, the F-statistic is the ratio of variation between sample means and variation within the samples.
Results table 1. I do not understand this table at all. The n numbers would suggest that not all rats were examined, with n=7 or n=6 stated for 3 of 4 groups, this is not mentioned in the methods section.
Answer: The seminiferous tubules were scored using the Johnsen score as shown in Table 1. The Johnsen score was explained in the material methods. Now in the material methods we added a part showing the number of subjects in the Johnsen score evaluation.
In the clozapine treated animals 4/6 are said to have Johnsen score of 8 and this is stated to be 77.7% (in text) and 78% in table. Surely 4/6 = 66.67% of rats.
Answer: our mistake thanks for noticing!
Results section Activity of Antioxidant Enzymes in Testicles and Prostate
What is GR?
Answer: Now we added abbreviation for all enzymes
Testicles: A decrease in catalase (CAT) activity (p<0.001) and an increase in coper-zink superoxide dismutase (SOD1) activity (p<0.05) were detected in clozapine-treated rats (Figure 4). Both ziprasidone and sertindole decreased both CAT activity (p<0.001) and selenium–glutathione peroxidase (GPx) activity (p<0.05) (Figure 4). Decreased glutathione reductase (GR) activity was detected only in ziprasidone-treated rats
Testicle results are reported before this section and should be moved. Graphical representation is said to be in figure 3 but is actually figure 4, with figures out of sync from here on. Results are not reported systematically, should be reported in more detail.
Answer: we aligned the order of the text with the order of the Figures shown in the manuscript
Thanks for spotting the misquotation of the Figures
For GPX what are the authors measuring? There are eight GPX proteins, which are expressed to differing degrees in different tissues e.g. GPX4 highly expressed in testes, but lower in prostate, GPX5 and 6 mostly expressed in epididymis. They are also seen if different subcellular compartments, some of which would be relevant to mitochondrial antioxidant protection others not.
Answer: We appreciate reviewer comment, we should have analyzed which GPx molecules were expressed in the testes and which in the prostates.
However the assay we used t-butyl hydroperoxide, after reaction with hydroperoxide moiety, both GPx1 and GPx4 are recycled by gathering reducing equivalents from glutathione, resulting in the formation GSSG.
GSSG -->GSH in presence of enzyme GR and NADPH --->NADP which we measure at 340nm.
Therefore we simultaneous measure both form of GPx
Reference:
Stolwijka M., Falls-Hubert C., Searby C., Wagner A., Buettner R. Simultaneous detection of the enzyme activities of GPx1 and GPx4 guide optimization of selenium in cell biological experiments Redox Biology 32, 2020, 101518.
Prostate results are figure 5 not figure 4.
Answer: Thanks, changed!
Negative correlation of prostate weight and SOD1/GR – how was this calculated? There is no sig diff in prostate weight noted between control group and clozapine or ziprasidone treated group and there is no sig diff in SOD1 and GR between control and sertindole so I find it hard to comprehend how this can be stated.
Answer: It was calculated by the method of Spearman's rank correlation coefficient using GraphPad Prism 8.0.2.
https://www.graphpad.com/guides/prism/latest/statistics/stat_interpreting_results_correlati.htm)
Correlation analysis (in this case Spearman's r) and Hypothesis testing (in this case analysis of variance - ANOVA) answer different questions. Correlation analysis is a statistical method used to measure the strength of the linear relationship between two variables (in our case prostate weight and SOD1/GR) and compute their association. On the other hand, hypothesis testing implies that a specific statement or hypothesis is generated about a population parameter, and sample statistics are used to assess the likelihood that the hypothesis is true. ANOVA is a test of hypothesis that is appropriate to compare means of a continuous variable in two or more independent comparison groups.
The authors state “The mean plasma prolactin and testosterone concentrations differed between treatment groups, but without statistical significance” however if there is not statistical significance this cannot be said to be so.
Answer: That has been changed:
There were no statistically significant changes in the mean plasma prolactin and testosterone concentrations between treatment groups, (Figure 6), probably due to large variability within the groups.
Does prolactin/testosterone ratio correlate with prostate weight? This would seem the most striking finding in sertindole group
Answer: We did that correlation and answer is No!
Could the authors test reproductive capacity in these rats? i.e did the changes in spermatogenesis reduce ability to father offspring?
Answer: Thanks for suggestion; we will include that for further experiments.
Discussion
Over interpretation of results, for example the assumption of increased hydrogen peroxide causing damage to the tissue where the authors have not shown that H2O2 is increased.
Answer: You are absolutely right, we did not measure H2O2, we only commented from the aspect of reducing the activity of the enzymes that remove it, so we mitigated our comments. That part has been rewrote:
Increased SOD1 activity most likely decreased superoxide in the cells whilst generating hydrogen peroxide. However, elimination of hydrogen peroxide was likely to be reduced due to the decreased activity of hydrogen peroxide decomposing enzymes (CAT and GPx). Decreased CAT and GPx activities directly correlated with the condition of the seminiferous tubules revealed by the Johnsen score suggesting that hydrogen peroxide accumulated in cells might be involved in testicular tissue damage.
Overall I am not convinced they show that ROS is what is causing the phenotype in these rats, further confirmatory studies/assays would need to be carried out. Longer term treatment of the rats perhaps, fertility studies and proof that ROS is, in fact, increased in these tissues.
Answer: We appreciate reviewer comment, in future studies we will incorporate your suggestions. However, here we measured antioxidant defence enzymes activity and found that ROS and oxidative stress have also role in pathological outcome. So, this is not ROS that causing that phenotype, but contribute.
Length of the treatment: comparisons of the life spans of rats and humans at different stages in their lives show that rats grow rapidly during their childhood and become sexually mature around the sixth week, but reach social maturity 5-6 months later. At adulthood, each animal day is roughly equivalent to 34.8 human days (ie, one rat month is comparable to three human years). Our rats were three months old, that means they are equivalent to sexually mature rats. Therefore 4-week treatment represents three-year of treatment. That duration for rats represents the long term treatment
Johnson or Johnsen, which is correct?
Answer: Thanks for noticing mistake Johnsen is correct!
Materials and methods
Is water an appropriate control? What is the formulation of the tablets? Presumably there are other components than just the drug which need to be accounted for in the control rats.
Answer: We followed protocols from already published manuscripts according for oral administration drugs in drinking water. One of them was:
Terry A. V. J R., Warner S.E.L., Vandenhuerk A. Pillai, S. P. Mahadik, G. Zhang, Bartlett M.G. Negative effects of chronic oral chlorpromazine and olanzapine treatment on the performance of tasks designed to assess spatial learning and working memory in rats Neuroscience 156 (2008) 1005–1016
For antioxidant assays was the tissue extract supernatant used without any further modifications?
Tissue extract supernatant for antioxidant assays was used without any further modifications.
Reviewer 2 Report
The Nikolić-Kokićet al., 2022, Manuscript ID: ijms- 1957647 addresses the aspects of redox homeostasis and hormonal homeostasis in antipsychotic drugs-mediated testicular changes. A search on Pubmed.gov for the terms "antipsychotic" and "testis" keywords resulted in only 34 hits.
There are few queries and few suggestion which makes this manuscript more representable to be publish.
1. With these results taken into consideration, it’s very hard to understand the mechanism of antipsychotic drugs on testicular changes and infertility? Therefore the authors has to perform some in-depth mechanistic approach to depict the underlying mechanism?
2. Why the authors have not included the infertility analysis mainly sperm analysis like sperm volume motility and viability. This gives the manuscript more strength and more interest to readers.
3. It will give more strength if the authors can include the apoptosis assays in the slide?
4. The results need to be authenticated utilizing protein level changes of apoptosis markers?
5. Androgen or testosterone can be systhesized through multiple organs such as adrenal gland so intra-testicular testosterone level need to be included?
6. The dose of the drugs need to be validated in there system by performing ALT AST level in the animals?
Author Response
The Nikolić-Kokić et al., 2022, Manuscript ID: ijms- 1957647 addresses the aspects of redox homeostasis and hormonal homeostasis in antipsychotic drugs-mediated testicular changes. A search on Pubmed.gov for the terms "antipsychotic" and "testis" keywords resulted in only 34 hits.
There are few queries and few suggestion which makes this manuscript more representable to be publish.
- With these results taken into consideration, it’s very hard to understand the mechanism of antipsychotic drugs on testicular changes and infertility? Therefore the authors has to perform some in-depth mechanistic approach to depict the underlying mechanism?
Answer: We agree, and appreciate Reviewers comment. The data regarding adverse effects of antipsychotic drugs on male reproductive system are still rare, and recent studies pointed to oxidative stress as one of the molecular mechanisms that could contribute to testicular degeneration and, on the long run, to infertility. We have to agree that, similarly to other studies, current study is rather descriptive. We demonstrated that atypical antipsychotics disturb redox homeostasis in the testis, and this was correlated to disturbed spermatogenesis. Yet, we have confirmed the proposed causative role of oxidative stress in drug-induced disruption of reproductive function in males. Evaluation of the exact mechanisms might include further assessment of the role of hydrogen peroxide. It is our intention to conduct such experiments in fresh tissue, as a part of future studies, since recent literature shows that sample storage duration alters the quantification of oxidative stress markers. (Costa et al., Storage Duration Affects the Quantification of Oxidative Stress Markers in the Gastrocnemius, Heart, and Brain of Mice Submitted to a Maximum Exercise. Biopreservation and Biobanking 2022; 20:3-11. https://doi.org/10.1089/bio.2020.0126).
- Why the authors have not included the infertility analysis mainly sperm analysis like sperm volume motility and viability. This gives the manuscript more strength and more interest to readers.
Answer: We agree with the Reviewer that additional infertility analysis would strengthen the current results, however we can only perform these experiments in our future studies, with new set of animals.
- It will give more strength if the authors can include the apoptosis assays in the slide?
- The results need to be authenticated utilizing protein level changes of apoptosis markers?
Answer: We are grateful for the Reviewers comments (3 and 4), and we agree that assessment of apoptosis would improve this manuscript, unfortunately 10 days is not enough time to perform all necessary measurements in order to fully characterize possible apoptosis in the testis. Nevertheless, since our tissue sections were stained with hematoxylin and eosin, we have evaluated the presence of apoptotic cells and pyknotic nuclei, changes preceding apoptosis, and focal necrosis in the testis. Therefore sentence has been incorporated in results:
Apoptosis as a form of programmed cell death was a rare finding only as a physiological. We didn't evident pathological apoptosis on routine testicular slides stained by hematoxylin eosin (HE).
- Androgen or testosterone can be systhesized through multiple organs such as adrenal gland so intra-testicular testosterone level need to be included?
Answer: We are grateful for these valuable suggestions, however at the moment we do not have tissue from this study kept, and we believe that determination of intra-testicular testosterone level in whole cell extract would not reflect actual situation in the tissue.
Measuring of testosterone/prolactin ratio is indicator of hormonal balance. In our experiment this balance is disturbed, suggesting hormonal imbalance. The idea of the work was not to monitor hormones and introduce the hormonal mechanisms of action of drugs, but how and in what direction they can affect the antioxidant defense, which is the topic of the work.
- The dose of the drugs need to be validated in there system by performing ALT AST level in the animals?
Answer: Dose of the drugs was validated but not presented in manuscript. Tested by one–way analysis of variance (ANOVA) followed by Tukey’s multiple comparison post-hoc test. ALT of Controls 13.8 ± 1.6; Clo 11.6±0.5; Zip 14.8±2.1; Ser 12.0 ± 2.1 as well as AST level: Controls 276.4 ± 21.5; Clo 244.8 ± 21.09; Zip 261.2 ± 29.2; Ser 258.2 ± 16.8. Since there was no differences between groups, therefore dose applied in our experiment were considered as safe.
Round 2
Reviewer 1 Report
The authors have adequately answered some of my minor points but the major criticisms, that they are measuring total GPX activity, have not shown increased ROS or decreased NADPH/NADH pools have not been addressed.
Original Reviewer comment For GPX what are the authors measuring? There are eight GPX proteins, which are expressed to differing degrees in different tissues e.g. GPX4 highly expressed in testes, but lower in prostate, GPX5 and 6 mostly expressed in epididymis. They are also seen if different subcellular compartments, some of which would be relevant to mitochondrial antioxidant protection others not.
Answer: We appreciate reviewer comment, we should have analyzed which GPx molecules were expressed in the testes and which in the prostates.
However the assay we used t-butyl hydroperoxide, after reaction with hydroperoxide moiety, both GPx1 and GPx4 are recycled by gathering reducing equivalents from glutathione, resulting in the formation GSSG.
GSSG -->GSH in presence of enzyme GR and NADPH --->NADP which we measure at 340nm.
Therefore we simultaneous measure both form of GPx
Reviewer follow up = all GPX activity will be measured by this
Original reviewer comment Over interpretation of results, for example the assumption of increased hydrogen peroxide causing damage to the tissue where the authors have not shown that H2O2 is increased.
Answer: You are absolutely right, we did not measure H2O2, we only commented from the aspect of reducing the activity of the enzymes that remove it, so we mitigated our comments. That part has been rewrote:
Increased SOD1 activity most likely decreased superoxide in the cells whilst generating hydrogen peroxide. However, elimination of hydrogen peroxide was likely to be reduced due to the decreased activity of hydrogen peroxide decomposing enzymes (CAT and GPx). Decreased CAT and GPx activities directly correlated with the condition of the seminiferous tubules revealed by the Johnsen score suggesting that hydrogen peroxide accumulated in cells might be involved in testicular tissue damage.
Reviewer follow up Throughout the long and rambling discussion there are many assumptions about increased ROS/decreased oxidative reserve BUT without proof of this the conclusions the authors make cannot be substantiated
Original reviewer comment Overall I am not convinced they show that ROS is what is causing the phenotype in these rats, further confirmatory studies/assays would need to be carried out. Longer term treatment of the rats perhaps, fertility studies and proof that ROS is, in fact, increased in these tissues.
Answer: We appreciate reviewer comment, in future studies we will incorporate your suggestions. However, here we measured antioxidant defence enzymes activity and found that ROS and oxidative stress have also role in pathological outcome. So, this is not ROS that causing that phenotype, but contribute.
Reviewer follow up The results from these studies need to be included in this manuscript for the authors to draw the conclusions they have
Author Response
Reviewer 1
The authors have adequately answered some of my minor points but the major criticisms, that they are measuring total GPX activity, have not shown increased ROS or decreased NADPH/NADH pools have not been addressed.
Original Reviewer comment For GPX what are the authors measuring? There are eight GPX proteins, which are expressed to differing degrees in different tissues e.g. GPX4 highly expressed in testes, but lower in prostate, GPX5 and 6 mostly expressed in epididymis. They are also seen if different subcellular compartments, some of which would be relevant to mitochondrial antioxidant protection others not.
Answer: We appreciate reviewer comment, we should have analyzed which GPx molecules were expressed in the testes and which in the prostates.
However the assay we used t-butyl hydroperoxide, after reaction with hydroperoxide moiety, both GPx1 and GPx4 are recycled by gathering reducing equivalents from glutathione, resulting in the formation GSSG.
GSSG -->GSH in presence of enzyme GR and NADPH --->NADP which we measure at 340nm.
Therefore we simultaneous measure both form of GPx
Reviewer follow up = all GPX activity will be measured by this
Answer: We appreciate reviewer comment, we maybe should have measured different GPx activities using different substrates utilized by individual GPx in the testes as well as in the prostates to better qualify their involvement including all discussion about cellular compartmentalization that again requires different preparation of tissues for analyses. However, there is no unique mode of action of atypical antipsychotics (APDs) to target precise intracellular oxidative pressure, and since they are in intensive therapeutic usage, without more detailed description on peripheral tissue activity we were focused on overall possibility of tissue oxidative stress and to measuring activity level of antioxidant enzymes that: 1) follow cellular oxidative state, and 2) enable as to conclude to some significant extent of the nature of oxidative processes induced by APDs. Therefore, our aim was to measure total GPxs enzymatic impact regarding peroxides metabolism. To achieved this we used t-butyl hydroperoxide in the assay, because after reaction with hydroperoxide moiety, both GPx1 and GPx4 are recycled by gathering reducing equivalents from glutathione, resulting in the formation GSSG.
GSSG -->GSH in presence of enzyme GR and NADPH --->NADP which we measure
at 340nm.
Therefore we simultaneous measure both form of GPx.
Reviewer follow up = all GPX activity will be measured by this
Original reviewer comment Over interpretation of results, for example the assumption of increased hydrogen peroxide causing damage to the tissue where the authors have not shown that H2O2 is increased.
Answer: You are absolutely right, we did not measure H2O2, we only commented from the aspect of reducing the activity of the enzymes that remove it, so we mitigated our comments. That part has been rewrote:
Increased SOD1 activity most likely decreased superoxide in the cells whilst generating hydrogen peroxide. However, elimination of hydrogen peroxide was likely to be reduced due to the decreased activity of hydrogen peroxide decomposing enzymes (CAT and GPx). Decreased CAT and GPx activities directly correlated with the condition of the seminiferous tubules revealed by the Johnsen score suggesting that hydrogen peroxide accumulated in cells might be involved in testicular tissue damage.
Measurement of hydrogen peroxide in tissues is methodological very difficult task that can gain very inconsistent results that cannot reflect situations in vivo. This can be explained and elaborated in let say additional review article. The same situations are for superoxide concentration measurements.
Reviewer follow up Throughout the long and rambling discussion there are many assumptions about increased ROS/decreased oxidative reserve BUT without proof of this the conclusions the authors make cannot be substantiated
Answer: To comply with Reviewer’s suggestion the discussion is now modified and shortened.
Original reviewer comment Overall I am not convinced they show that ROS is what is causing the phenotype in these rats, further confirmatory studies/assays would need to be carried out. Longer term treatment of the rats perhaps, fertility studies and proof that ROS is, in fact, increased in these tissues.
Answer: We appreciate reviewer comment, in future studies we will incorporate your suggestions. However, here we measured antioxidant defence enzymes activity and found that ROS and oxidative stress have also role in pathological outcome. So, this is not ROS that causing that phenotype, but contribute.
Reviewer follow up The results from these studies need to be included in this manuscript for the authors to draw the conclusions they have
Answer: We appreciate the suggestion but this represents a completely new study.
Reviewer 2 Report
The authors tried to justify the comments and queries raised by me.
I suggested authors few more queries and changes for the improvement of manuscript.
1. Taking consideration of the authors tissue limitation, the authors are requested to calculate the germinal epithelium height and seminiferous tubule diameter (you can follow the article doi:10.1016/j.ygcen.2018.06.002) and if possible follow and cite the article (https://doi.org/10.1016/j.bbadis.2018.11.019, doi: 10.1016/j.biochi.2019.10.014) to count the advanced spermatocyte (preleptotene (PL), pachytene (P) and dividing spermatocytes (S) in different treatment groups.
2. Kindly include the ALT, AST data as tabular form and check the values and dilution as AST for control cannot be more than 50 U/L.
3. Do the authors checked the combined dose of all three drugs? If yes can you eloborate the results.
Author Response
Reviewer 2
The authors tried to justify the comments and queries raised by me.
I suggested authors few more queries and changes for the improvement of manuscript.
- Taking consideration of the authors tissue limitation, the authors are requested to calculate the germinal epithelium height and seminiferous tubule diameter (you can follow the article doi:10.1016/j.ygcen.2018.06.002) and if possible follow and cite the article (https://doi.org/10.1016/j.bbadis.2018.11.019, doi: 10.1016/j.biochi.2019.10.014) to count the advanced spermatocyte (preleptotene (PL), pachytene (P) and dividing spermatocytes (S) in different treatment groups.
Answer: We are grateful for the Reviewers’ useful comment, and the papers and methods suggested (which are now included in the second paragraph of the Discussion as ref 24 and 25).
However, in this study we did not use any morphometric analysis for seminiferous tubules. Our plan was to determinate it in the future studies, as the person responsible for histopathological analysis is out of the country, thus we cannot perform the measurements in a time provided for this revision (5 days).
For testicular biopsy interpretation, we used the criteria formulated by Johnsen (Johnsen SG. Testicular biopsy score count - a method for registration of spermatogenesis in human testes: normal values and results in 335 hypogonadal males. Hormones 1970;1:2-25.), which is known to have good reproducibility. Johnsen scores use a ten-point scoring system for quantifying spermatogenesis according to the profile of the cells encountered along the seminiferous tubules.
The seminiferous tubules were graded using the Johnsen score as: score 10 (complete spermatogenesis with regular tubules and many spermatozoa); score 9 (slightly impaired spermatogenesis with many late spermatids, disorganised epithelium); score 8 (Less than five spermatozoa per tubule, few late spermatids); score 7 (no late spermatids, many early spermatids); score 6 (no late spermatids, few early spermatids); score 5 (no spermatids, many spermatocytes); score 4 (no spermatids, few spermatocytes); score 3 (presence of spermatogonia only); score 2 (presence of Sertoli’s cells only); or score 1 (no seminiferous epithelium).
- Kindly include the ALT, AST data as tabular form and check the values and dilution as AST for control cannot be more than 50 U/L.
Answer: The results are now presented in Table1. We are grateful to the Reviewer for raising this question. We have consulted our veterinary responsible for animals in our facility. He advised us to perform additional analysis using fresh blood from new animals, so we prepared plasma and serum from control male Wistar rats (see the results below). Importantly for this study, our veterinary confirmed that there were no significant differences between control and treated groups, and he also commented that, based on his experience, the values for ALT and AST activities might differ depending whether they were determined using automatic analyzer or commercial kit (even the kits from different manufacturers might differ in range).
Our new results done on October 26. show that AST value is around 250 U/L in serum and 340 in plasma; and ALT around 75 U/L (now measured in the commercial laboratory using Biosystems kits 21531 for AST and 21533 for ALT) (the reference values are for humans).
Plasma
Serum
In addition, the most recently published data on ALT and AST activity in the plasma of Wistar rats from our facility are given below:
Đurašević et al., The Effects of a Meldonium Pre-Treatment on the Course of the Faecal-Induced Sepsis in Rats. IJMS 2021 Sep 8;22(18):9698. doi: 10.3390/ijms22189698.
Male Sprague-Dawley rats
Activities of ALT and AST activity in serum were measured by Roche Cobas C501 automated analyser (Roche Diagnostics, Mannheim, Germany), using ALTL and ASTL reagent cassette.
Kovacevic et al., Fructose-Rich Diet Attenuates Stress-Induced Metabolic Disturbances in the Liver of Adult Female Rats Journal of Nutrition, 2021 Dec 3;151(12):3661-3670. doi: 10.1093/jn/nxab294.
Female Wistar rats
Plasma ALT was determined by ALT assay (OSR6107; Beckman Coulter), whereas the concentration of plasma AST was determined by AST assay (OSR6109; Beckman Coulter). Analyses were performed on an Olympus AU400 Chemistry Analyzer (Olympus and Beckman Coulter) in a commercial laboratory (VetLab, Belgrade, Serbia).
- Do the authors checked the combined dose of all three drugs? If yes can you eloborate the results.
Answer: We appreciate the suggestion, but we have not investigated the effects of combined doses of these drugs.